# Monitoring of *Staphylococcus epidermidis* biofilm formation on platelet storage bag surfaces

Jolianne Matte[1,2], Sahra Fonseca[1], Jonathan Robidoux [1], Steve J. Charette[2,3], Marie-Pierre Cayer [1]*, Danny Brouard[1,2]*

1 Héma-Québec, Affaires Médicales et Innovation, Quebec City, QC, Canada, 2 Département de Biochimie, de Microbiologie et de Bio-Informatique, Université Laval, Quebec City, QC, Canada, 3 Institut de Biologie Intégrative et des Systèmes, Université Laval, Quebec City, QC, Canada

* marie-pierre.cayer@hema-quebec.qc.ca (MPC); danny.brouard@hema-quebec.qc.ca (DB)

## Abstract

Platelet concentrates (PCs) are stored at 20–24˚C in a biologically favorable environment that may support bacterial growth. *Staphylococcus epidermidis*, a typical contaminant, can form biofilms in PCs, complicating detection and increasing the risk of transfusion-transmitted bacterial infections. The material composition and surface texture of PC storage bags may influence biofilm formation. The impact of different PC storage bag materials on *S. epidermidis* biofilm formation was evaluated using the ISO 4768:2023(E) crystal violet (CV) assay. Four surface conditions were tested: polyvinyl chloride (PVC) plasticized with n-butyryl-tri(n-hexyl)-citrate (BTHC) – both smooth and rough sides, PVC plasticized with tri-(2-ethylhexyl)-trimellitate (TEHTM) and ethylene-vinyl acetate (EVA). Coupons and bags made from each material were used in the experiments. Biofilm-positive *S. epidermidis* was cultured in tryptic soy broth (TSB), PCs and plasma and added on plastic coupons under static conditions or directly in the bags with agitation. Bacterial enumeration and CV assay were performed on days 2, 5, and 7. In TSB, EVA coupons significantly formed more biofilm than the smooth side of PVC-BTHC or TEHTM over seven days. In PCs, more biofilm formed on the rough side of PVC-BTHC coupons than the smooth side, with no other differences between plastics, suggesting similar biofilm amount across PC bag materials in the presence of platelets. No biofilm was detected on coupons in plasma. Under continuous agitation and reduced oxygen levels, only the rough side of PVC-BTHC showed significant biofilm formation in TSB in PC storage bags over seven days. These findings highlight the need for standardized biofilm testing and suggest that some plastics are more conducive to biofilm formation under static conditions. However, during blood bank storage (i.e., continuous agitation and reduced oxygen levels), biofilm formation is limited, regardless of the platelet bag material, thereby reducing the risk of undetected bacterial contamination.

**Data availability statement:** All relevant data are within the manuscript and its Supporting information files.

**Funding:** Initials of the authors who received each award: JM Grant numbers awarded to each author: IT38130. The full name of each funder: MITACS Accelerate URL of each funder website: https://www.mitacs.ca/our-programs/accelerate/ Did the sponsors or funders play any role in the study design, data collection and analysis, decision to publish, or preparation of the manuscript? NO - The funders had no role in study design, data collection and analysis, decision to publish, or preparation of the manuscript.

**Competing interests:** The authors have declared that no competing interests exist.

**Abbreviations:** ATCC, American Type Culture Collection; AUC, Area under curve; BTHC, n-butyryl-tri(n-hexyl)-citrate; CFU, Colony-forming units; CV, Crystal violet; EVA, Ethylene-vinyl acetate; OD, Optical density; ODc, Optical density cut-off; PAS, Platelet additive solution; PCs, Platelet concentrates; PRTs, Pathogen reduction technologies; PVC, Polyvinyl chloride; Rpm, Rotations per minute; SD, Standard deviation; TEHTM, Tri-(2-ethylhexyl)-trimellitate; TSB, Tryptic soy broth; TTBIs, Transfusion-transmitted bacterial infections; WBD, Whole blood donation.

## Introduction

The storage conditions for platelet concentrates (PCs) in blood services—20–24°C under constant agitation—create an environment conducive to bacterial proliferation. Consequently, blood services systematically test all PCs prior to their clinical use [1–3]. The primary source of PC contamination is human skin bacteria, which can enter the donation through the collection needle during venipuncture [2–5]. *Staphylococcus epidermidis* is the staphylococcal species most frequently isolated from contaminated PCs [6–8].

To enhance transfusion safety, blood services in Canada and elsewhere have implemented preventive measures to mitigate contamination risks, particularly in PCs. These measures include skin disinfection before venipuncture together with the diversion of the first few milliliters of blood. More recently, automated culture detection systems and pathogen reduction technologies (PRTs) have been added [9,10]. While both strategies have significantly improved transfusion safety, they address bacterial contamination through distinct mechanisms and face different limitations. Culture-based methods may fail to detect slow-growing or biofilm-forming bacteria such as *S. epidermidis*, leading to false-negative results [11–14]. In contrast, PRTs aim to inactivate a broad range of pathogens, including bacteria, viruses and parasites. Their efficacy can be influenced by factors such as the initial bacterial load, the timing of application, and the resistance of certain species, including those capable of forming biofilms [15]. Moreover, rare cases of transfusion-transmitted bacterial infections (TTBIs) have been reported following PRT implementation, often due to post-treatment contamination or limitations in the inactivation process [16,17].

Despite these preventive strategies, one of the main reasons *S. epidermidis* poses a residual risk in transfusion safety is its ability to adhere to the inner surfaces of PC storage bags, where attachment can be boosted by plasma factors in PCs [18–20]. These biofilms, especially in the early formation stages, may escape detection during routine bacterial testing, thereby increasing the risk of TTBIs [21].

Biofilm formation may be influenced by the types of materials used to manufacture PC storage bags. PC containers are primarily composed of polyvinyl chloride (PVC), which requires approximately 40% plasticizers, such as n-butyryl-tri(n-hexyl)-citrate (BTHC) or tri-(2-ethylhexyl)-trimellitate (TEHTM or TOTM), to ensure properties like texture, thickness, gas permeability and flexibility [20,22,23]. The TEHTM plasticizer offers excellent resistance to heat and plasticizer migration, while BTHC is recognized for its low toxicity and biological compatibility. Plasticizer-free alternatives, such as polyolefin and its copolymer derivative, ethylene-vinyl acetate (EVA), may exhibit an irregular texture [22] prone to plasma proteins and platelets adsorption [24] that may promote bacterial biofilm formation. The main purpose of adding diamond-like (PVC-BTHC and PVC-TEHTM) or taffetas (EVA) textures to the plastic bag design is to prevent fusion of the inner surfaces during the sterilization process. EVA bags offer higher oxygen permeability than PVC bags [22] and allow better penetration of ultraviolet rays [25], which are used, for example, in PRTs.

Quantifying biofilm formation is commonly achieved through colorimetric assays including the crystal violet (CV) assay [26]. Even though previous studies have

investigated biofilm formation on various plastics used in PC storage bags, such as PVC with different plasticizers [19,20], significant protocol differences, particularly in strain preparation and selection, staining conditions and absorbance wavelength measurements [26,27], make it difficult to compare results between studies. The introduction of ISO 4768:2023(E), *Measurement method of anti-biofilm activity on plastic and other non-porous surfaces*, has standardized the CV assay by providing clear guidelines that facilitate the assessment and comparison of biofilm formation across various materials and conditions. This standard, published in July 2023, is the first of its kind, marking a significant step towards more reliable and reproducible testing methods for biofilm quantification.

Based on ISO 4768:2023(E), we assessed the impact of the composition and surface texture of primary materials used in PC storage bags on *S. epidermidis* biofilm formation, specifically focusing on PVC-BTHC (rough and smooth surfaces), PVC-TEHTM and EVA, the latter of which has not been studied in the context of biofilm formation. These plastics were chosen because they meet the required standards, with PVC-BTHC and EVA being the primary materials used in Canadian blood banks for PC storage.

## Materials and methods

### Blood products preparation

This study was approved by Héma-Québec's ethics committee (approval number: CER-2022–013). Plasma and PCs were prepared from whole blood donations (WBD) according to Héma-Québec's standard procedures. Briefly, WBD were obtained from healthy donors, recruited between 11/12/2023–19/03/2024. Each participant provided written informed consent. Reveos collection kits (TerumoBCT) were used to collect all 450 ml donations. WBD were stored for 16–24 hours at room temperature before being processed using the overnight, three components (3C) Reveos standard protocol, as previously described [28]. Compatible ABO interim platelet units were pooled, and leukoreduced using the Reveos pooling set (TerumoBCT) to create the final PCs. All PCs used in this study were suspended in 100% plasma and underwent routine screening according to our standardized operational practices. Platelet counts were performed before inoculation on a hematology system (Beckman Coulter, Coulter AcT™ 5diff AL) and sterility was confirmed before bacterial inoculation (bioMérieux, BacT/ALERT 3D).

### PC bag plastics

Four types of PC storage bags were evaluated: PVC-BTHC featuring one smooth side (S-PVC-BTHC) and one rough side (R-PVC-BTHC), PVC-TEHTM and EVA. The PC bag surfaces were analyzed using scanning electron microscopy (JEOL, JSM-6360 LV) at 22X magnification. The samples were mounted on conductive supports and metallized with gold using sputter coating.

### Bacterial strains and growth conditions

*S. epidermidis* (American Type Culture Collection [ATCC], strain 35984) was used in all experiments. Strain ATCC 35984 was selected to comply with the ISO 4768:2023(E) standardization [29] and because it has been widely used in previous biofilm related studies [6,30,31]. Consequently, this strain was considered a model strain for biofilm formation. *S. epidermidis* ATCC 12228 was used as a non-biofilm-forming strain and served as negative control [6,30,32,33].

The viability of *S. epidermidis* biofilm was tested in three distinct growth environments: tryptic soy broth (TSB), PCs and plasma. TSB (BD, cat. 211825), which contains 0.25% of glucose, served as a baseline comparison against the more complex matrices of blood products. Under the experimental conditions used in this study, *S. epidermidis* ATCC 35984 consistently formed biofilms in TSB without additional glucose supplementation.

Experiments were conducted at 20–24 °C under two conditions: (i) in static plates using 2.5×2.5 cm coupons cut from PC bags, and (ii) in PC storage bags under agitation, mimicking blood bank storage conditions. All tests described in the

following sections were performed on day 2, 5, and 7 after inoculation. For each condition, experiments were conducted in technical triplicates and repeated independently three times (i.e., three biological replicates; n = 3).

## Preparation of plastic coupons

The PC storage bags were cut in square coupons of 2.5 cm × 2.5 cm, which were decontaminated in 70% ethanol and washed twice with sterile 1X phosphate-buffered saline, as previously described [28], to eliminate any residual ethanol. Coupons were affixed to the bottom of sterile six-well polystyrene plates using Steri-Strips (Cardinal Health, cat. 3MR1540).

## Preparation of inoculum in growth medium

An agar plate was inoculated from a frozen culture of *S. epidermidis* ATCC 35984 and incubated at 37 ± 1°C for 24 hours. A fresh culture was then obtained by inoculating 10 ml of TSB with a few colonies from the agar plate, followed by overnight incubation at 37 ± 1°C with continuous agitation at 250 rotations per minute (rpm) (Thermo Scientific, MaxQ 4000 orbital shaker). Fresh cultures were used instead of frozen stocks to ensure physiological consistency and maintain bacteria in the exponential growth phase, critical for initiating robust and reproducible biofilm formation.

The bacterial concentration was estimated by spectrophotometry (Thermo Scientific, Genesys 10S UV-Vis) at 600 nm and adjusted to an estimated $10^4$ colony-forming units (CFU)/ml, in accordance with the ISO 4768:2023(E) standard. To validate this estimation, colony counts were performed on the adjusted suspension under all experimental conditions (TSB, plasma or PCs).

This bacterial load was chosen to allow for the observation of changes in culture viability over time. A higher load could mask variations in biofilm formation, while a lower load might prevent observing viability loss. Variations in the initial bacterial load (± 1 log) may have occurred due to the inherent limitations of spectrophotometric estimation, particularly when working with *S. epidermidis* ATCC 35984. This strong biofilm-forming strain tends to form aggregates, complicating resuspension and thereby affecting the accuracy of optical density measurements.

## Inoculation of bacteria for coupons and PC storage bags

A 5 ml volume of the *S. epidermidis* suspension, adjusted to ~$10^4$ CFU/ml in the corresponding test medium (TSB, plasma, PCs) was added to each well containing plastic coupons to ensure full surface coverage. The plates were then incubated at 20–24 °C with ≥ 90% relative humidity for seven days without agitation. The same bacterial suspension (~$10^4$ CFU/ml in TSB, plasma, or PCs) was also introduced into the four types of bags. For each condition, the bags were filled to reach a surface/volume ratio of 2 cm²/ml, calculated based on one side of the bag. This value reflects the typical exposure ratio found in an average PC. For TSB and plasma experiments, the bags were filled directly. For PC experiments, the bags were first sealed using a plastic welder (Stapler Warehouse, KF-300H) to reduce internal surface area and minimize the volume of PCs required. A schematic representation of the PC storage bags used in this study, including their dimensions and the volumes added under each experimental condition, is shown in S1 Fig. The bags were incubated at 20–24 °C on a platelet agitator (70 ± 1 cycles per minute with a 3.8 cm amplitude) for seven days.

## Crystal violet assay

The CV assay was conducted following the adapted ISO 4768:2023(E) [29]. For the coupons analysis, the liquid was first removed from the plates. The Steri-Strips were then removed, and the plastic coupons were washed in sterile water and transferred to a new sterile six-well plate. A 5 ml volume of filtered 0.1% CV, prepared from a commercially available 0.3% CV solution (BD, cat. 212525), was added to each well. The plate was incubated at room temperature for 30 minutes. After incubation, the coupons were immersed in sterile water and washed by gently moving them back and forth about five

times before being transferred to a new sterile plate to minimize potential bacterial attachment to the original plate surface. Only the biofilm formed on the coupons was analyzed and blank controls were included to account for non-specific adhesion. The CV was eluted from the biofilm by adding 10 ml of a 1:4 acetone/ethanol solution to each well. The plate was then agitated at 100 rpm for 15 minutes. The contents of each well were transferred in triplicate into a 96-well plate and absorbance was read at a maximum wavelength of 590 nm using a microplate reader (BioTek, Synergy H1). All optical density (OD) values are presented as mean ± standard deviation (SD) across the three replicates (n).

For the four biofilm growth experiments in PC bags, TSB or PC content was carefully removed by bag inversion after incubation. Three 2.5 cm × 2.5 cm plastic coupons were randomly cut from the bottom plastic surface using sterile scissors on days 2, 5, and 7 of incubation (one bag for each incubation period). The coupons were washed in sterile water and placed in a sterile six-well plate. The following steps of the analytical procedure were the same as described above for coupons. Parallel experiments were performed in medium without bacterial inoculation as a control, corresponding to blank samples for both coupons and PC bags experiments.

The blank OD values were subtracted from the test sample values to eliminate background signal from the media and non-specific platelet adhesion to the plastics. Negative controls were performed using the same procedure, but with *S. epidermidis* ATCC 12228, a biofilm-negative strain.

The detection threshold for biofilm formation was defined using the method described by Stepanović et al. [34]. In summary, the OD cut-off (ODc) value was determined by adding three times the SD of the negative control to its mean OD: ODc = Mean OD of negative control + 3 × SD of negative control. The calculations were performed using *S. epidermidis* ATCC 12228 with the R-PVC-BTHC in TSB on day 7 based on three independent experiences for both coupons and PC bags. This experimental condition was selected to establish the detection threshold for biofilm formation, as it was known to promote strong biofilm growth for strain ATCC 35984.

### Quantification of bacterial viability by colony count

For PC bags, an aliquot of the growth medium was collected on each day of analysis to quantify planktonic bacterial viability only, as biofilm sampling have disrupted the system. For coupons, the entire growth medium in each well was transferred into a tube along with its corresponding plastic coupon. The tube was vortexed for ten seconds to dislodge both planktonic and surface-adhered bacteria, allowing quantification of the total bacterial population.

The resulting suspensions were serially diluted and plated on plate count agar. The inoculated agar plates were then incubated at 37 ± 1 °C for 16–24 hours and the number of colonies was visually counted. All bacterial concentrations are presented as mean log CFU/ml (n = 3).

### Statistical analysis

The Student's two-sample t-test for mean differences (α = 0.05) was used to compare biofilm formation between experimental conditions at day 2, 5, and 7. The areas under the curves (AUC) of the observed profiles were calculated to compare the global behavior of the plastics in terms of biofilm formation over the seven-day storage period. Mean values were compared using the Student's t-test for mean differences (α = 0.05). When applicable, effect sizes (Cohen's *d*) were calculated to assess the magnitude of differences between groups. The statistical analyses were conducted with GraphPad Prism 10, version 10.2.3.

## Results

### Surface textures of PC storage bag plastics

Microscopy images were captured to examine the surface morphology of plastic materials used in PC storage bags. At 22X magnification, the detailed texture of each plastic was observed. Both R-PVC-BTHC and PVC-TEHTM exhibited

diamond-patterned textures (one diamond: 400 µm in width and 800 µm in length). In contrast, S-PVC-BTHC exhibited a relatively smoother surface, whereas EVA displayed an irregular, uneven texture with cracks and small protrusions (Fig 1).

**Rough PVC-BTHC and EVA coupons favoring biofilm formation in tryptic soy broth**

The ODc was calculated using the biofilm-negative strain *S. epidermidis* ATCC 12228 in TSB on R-PVC-BTHC static coupons incubated seven days at 20–24°C and fixed at an OD of 0.6. Considering the ODc, no biofilm formation was detected on day 2 for all plastics and over the seven-day incubation for S-PVC-BTHC (OD ≤ 0.6) (Fig 2A). Biofilm forma-tion reached detectable levels by day 5 for R-PVC-BTHC and EVA and by day 7 for PVC-TEHTM. A significant difference in biofilm quantity was observed on day 5 between EVA (OD = 1.7 ± 0.4) and S-PVC-BTHC (OD = 0.4 ± 0.3; $p < 0.01$). On day 7, although this difference appears similar, no statistically significant difference was detected ($p = 0.14$), mainly due to the large standard deviation. However, a large effect size was observed ($d = 1.20$) between EVA and S-PVC-BTHC, suggesting a potentially meaningful difference in biofilm formation that may warrant further investigation. Overall, biofilm formation behavior remained consistent between plastic samples on day 7. Bacterial viability was comparable across all plastics throughout the storage period, with a final load of ~8.5 log CFU/ml (Fig 2B).

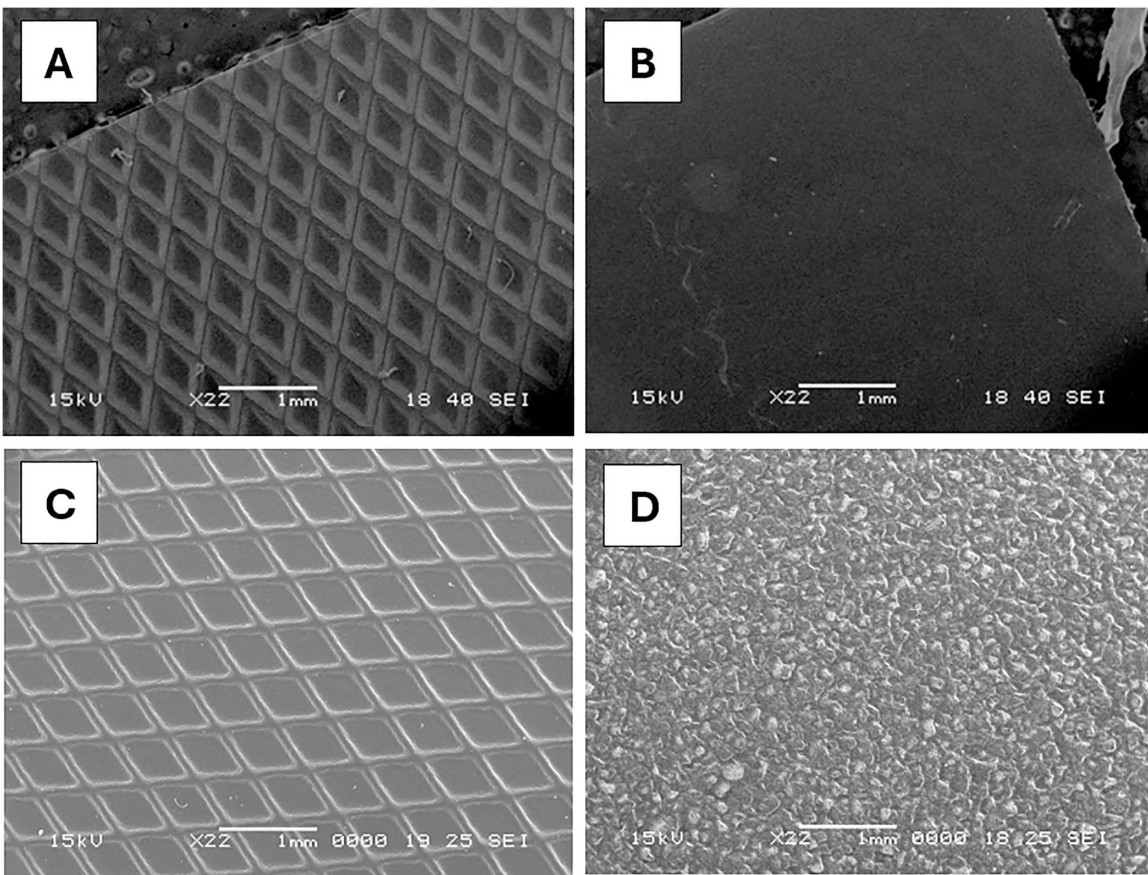

**Fig 1. Microscopic images of the tested PC bags.** Scanning electron microscopy was used on the rough side (A) and the smooth side (B) of the PVC-BTHC bag, PVC-TEHTM bag (C) and EVA bag (D).

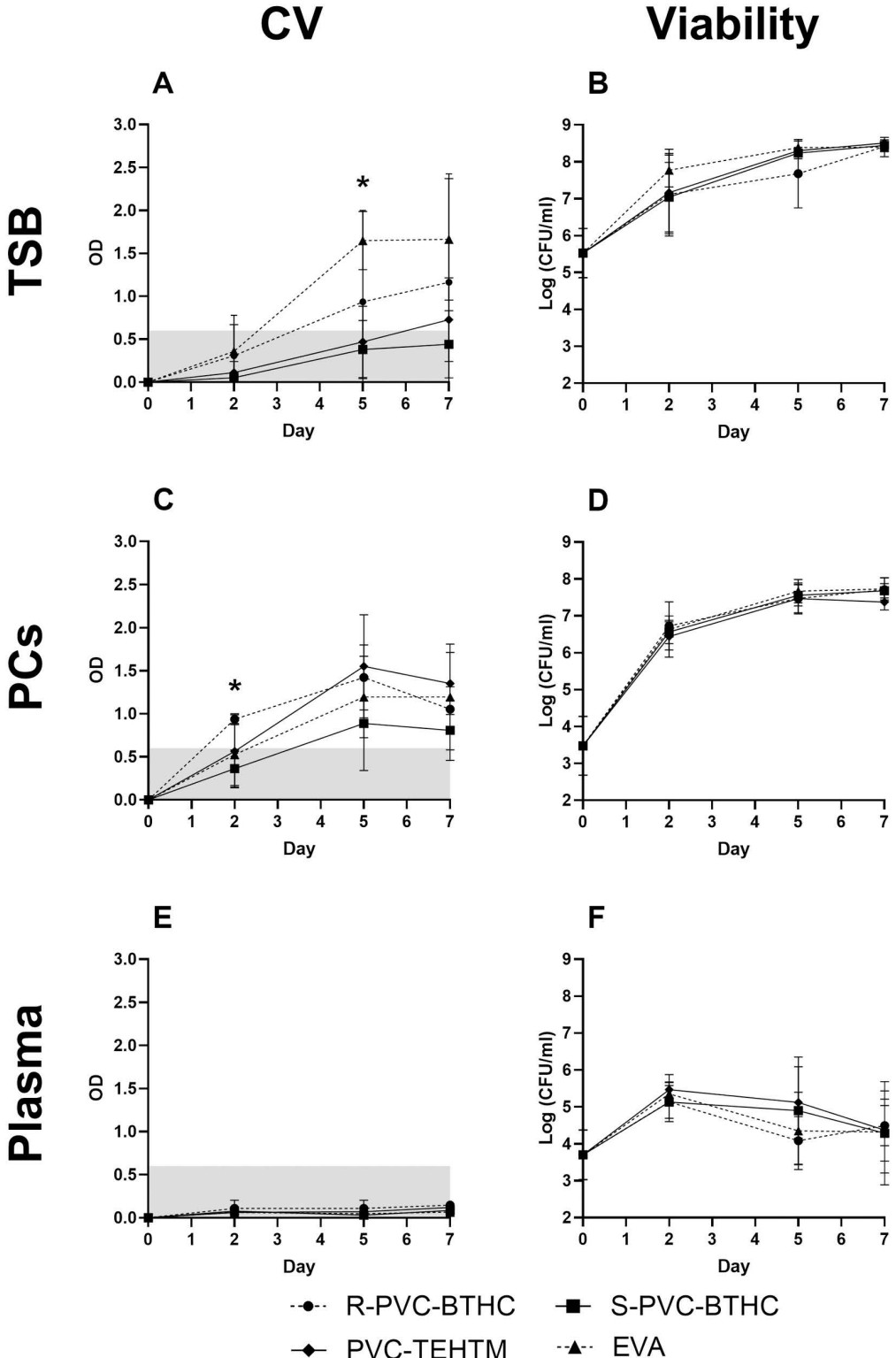

**Fig 2. Biofilm formation and viability of *S. epidermidis* ATCC 35984 on plastic bag coupons.** Mean OD by CV assay (A, C, E) and mean total bacterial concentration (log CFU/ml) (B, D, F) are shown for day 2, 5, and 7 of incubation in TSB (A, B), PCs (C, D) and plasma (E, F) (n = 3). The shaded gray area represents the threshold for biofilm formation. Viability values represent the total bacterial population recovered from both the medium and the

coupon surface (planktonic + biofilm-associated cells). Biofilm was assayed on R-PVC-BTHC (••●••), S-PVC-BTHC (■), PVC-TEHTM (◆) and EVA (••▲••) are represented. Statistically significant differences were observed between R-PVC-BTHC and S-PVC-BTHC in PCs on day 2 (A, *$p < 0.05$) and between EVA and S-PVC-BTHC in TSB on day 5 (C, *$p < 0.05$). Error bars represent the standard deviation across three biological replicates.

### Consistent biofilm formation across plastics coupons with presence of platelets

Considering the calculated ODc of 0.6 for TSB coupons, biofilm formation was observed from day 2 for R-PVC-BTHC and from day 5 for EVA, PVC-TEHTM, and S-PVC-BTHC (Fig 2C). Specifically, on day 2, R-PVC-BTHC promotes faster biofilm formation, showing a significant difference between the rough (OD = 0.9 ± 0.3) and smooth sides (OD = 0.4 ± 0.2) ($p < 0.01$). However, the final biomass quantity is comparable to other conditions. Indeed, the presence of platelets tends to normalize biofilm formation across all tested plastics for days 5 and 7. Despite no significant differences, PVC-TEHTM showed the highest biofilm formation (OD = 1.4 ± 0.5), while S-PVC-BTHC showed the lowest (OD = 0.8 ± 0.5; $p = 0.14$) at the end of incubation. Bacterial growth on all plastics remained consistent throughout all days, with a final growth of ~7.5 log CFU/ml (Fig 2D).

### Absence of biofilm formation by *S. epidermidis* in plasma

No biofilm was observed when *S. epidermidis* was cultured in plasma on coupons, regardless of the type of plastic tested or the time points considered (days 2, 5, and 7) (Fig 2E). Conversely, in TSB and PCs, most observed absorbance values consistently exceeded the ODc of 0.6 on days 5 and 7. However, the OD values in plasma never surpassed the detection threshold; the maximum biofilm OD value (OD = 0.13) was observed on day 7 for R-PVC-BTHC. Additionally, bacterial growth was significantly lower in plasma compared to other growth media, peaking at 5.5 log CFU/ml on day 2 with PVC-TEHTM and decreasing thereafter (Fig 2F).

### Biofilm formation from coupons to PC storage bags

To quantify biofilm formation in PC storage bags, we adapted the ISO 4768:2023(E) standard protocol, which typically suggests the use of plastic coupons under static conditions. Sections of the complete bags were randomly cut after incubation to fit protocol requirements, replicating blood bank storage conditions ([20–24 °C], under constant agitation). The ODc was calculated using the biofilm-negative *S. epidermidis* strain ATCC 12228 and set at 0.1. No biofilm was detected in TSB, regardless of the type of plastic or the day of analysis (Fig 3A), despite a final bacterial concentration of ~8.2 log CFU/ml for all plastic samples (Fig 3B). In PCs, biofilm was detected only with the R-PVC-BTHC condition on day 5, with a maximum OD of 0.1 (Fig 3C). Aggregates in suspension indicated loosely adhered biofilm, which may have been removed during washing and consequently not quantified. Despite the small amount of biofilm, bacterial growth remained consistent, bacterial load being ~7.0 log CFU/ml across all materials on day 7 (Fig 3D). Plasma was not tested directly in the PC bags, as no biofilm was observed on the plastic coupons.

### Impact of growth media and incubation parameters on biofilm formation

Biofilm formation on coupons between TSB and PCs was compared for each type of plastic and each day of analysis; however, no statistical differences were observed ($p > 0.05$). To further explore the cumulative biofilm formation behavior over the entire incubation period, we calculated the AUC for each condition (Fig 4). While individual time-point comparisons provide insight into the kinetics of biofilm development, AUC offers a global measure of biofilm accumulation, reducing the impact of variability at specific time points. This integrative approach revealed that EVA had a significantly higher cumulative biofilm formation in TSB compared to S-PVC-BTHC and PVC-TEHTM ($p < 0.01$; $d = 3.68$) (Fig 4A), and that R-PVC-BTHC accumulated more biofilm than S-PVC-BTHC in PCs ($p < 0.05$; $d = 1.96$) (Fig 4B). When comparing the AUC

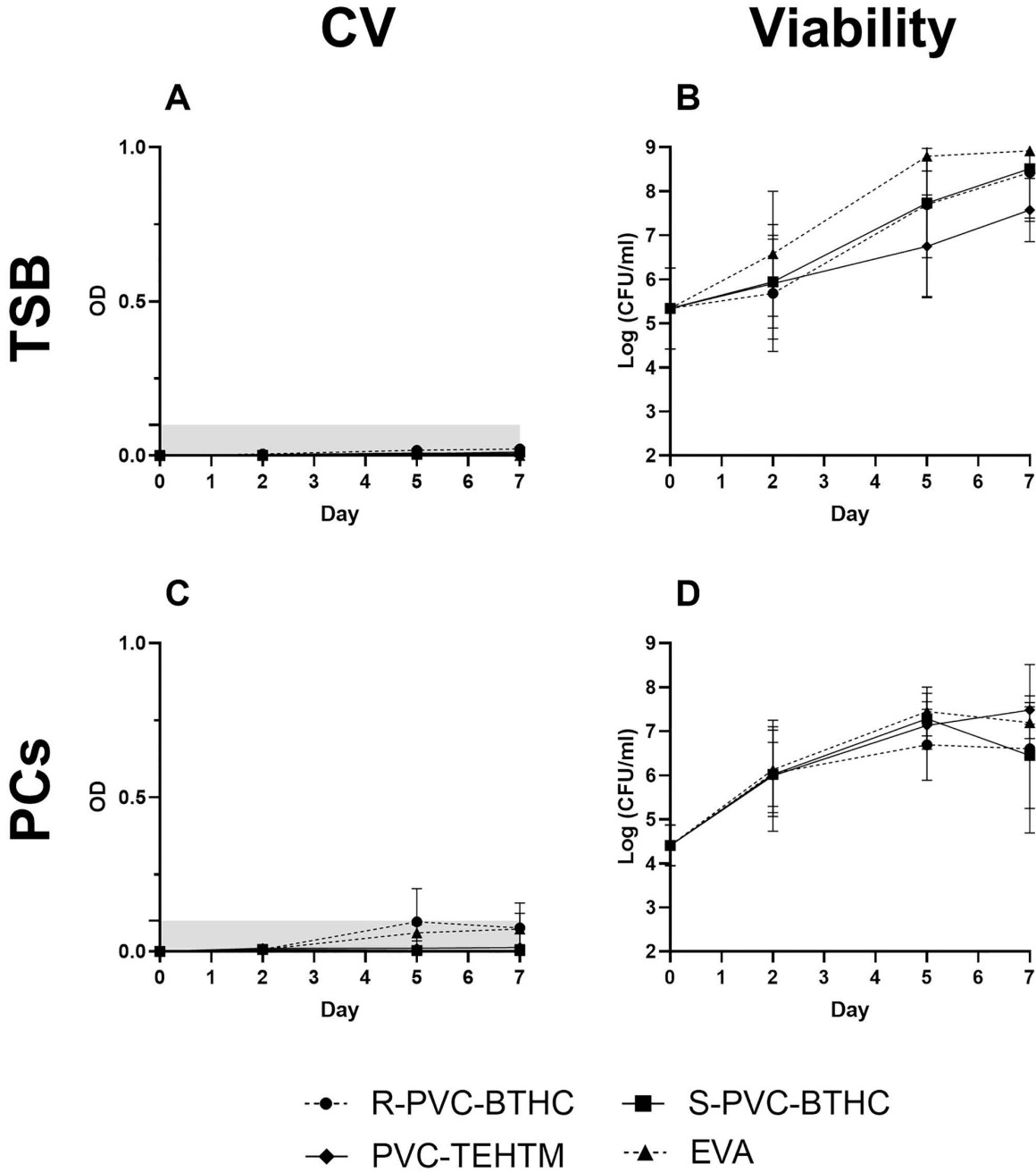

**Fig 3. Biofilm formation and viability of *S. epidermidis* ATCC 35984 in PC storage bags.** Mean OD by CV assay (A, C) and mean planktonic bacterial concentration (log CFU/ml) (B, D) are shown for day 2, 5, and 7 after incubation in TSB (A, B) and PCs (C, D) (n = 3). Viability values represent planktonic cells only, as biofilm sampling was not performed in bags to preserve system integrity. Biofilm was assayed on R-PVC-BTHC ··●··), S-PVC-BTHC (■), PVC-TEHTM (◆) and EVA (··▲··) are represented. No statistically significant differences were observed. Error bars represent the standard deviation across three biological replicates (*p* > 0.05).

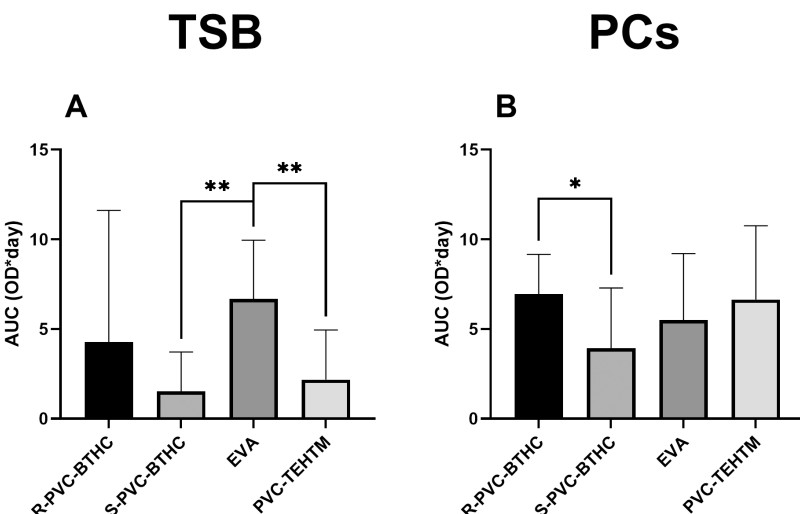

**Fig 4. Cumulative biofilm formation of *S. epidermidis* ATCC 35984 on plastic bag coupons, expressed as AUC.** AUC values were calculated from CV assay absorbance profiles over the seven-day incubation period in TSB (A) and PCs (B). Statistically significant differences were observed between EVA and S-PVC-BTHC and PVC-TEHTM in TSB (\*\**p* < 0.01) and between R-PVC-BTHC and S-PVC-BTHC in PCs (\**p* < 0.05). Error bars represent the standard deviation across three biological replicates.

between TSB and PCs, a trend emerges indicating that biofilm levels are generally higher in PCs than in TSB, except for EVA. Statistically, only the AUC calculated from the PVC-TEHTM growth profiles indicated a significantly higher amount of biofilm in PCs (*p* < 0.05). These findings support the trends observed in the time-point data and reinforce the influence of surface texture and plastic type on biofilm development.

Based on these results, additional experiments were conducted to further investigate the effects of agitation and gas permeability in PC bags (S1 File). For the agitation condition of the coupons, no biofilm formation was observed (S2 Fig). Similarly, no biofilm was detected in PC bags when incubated without agitation. Also, the impact of initial bacterial load in static PC bags was evaluated by inoculating them with an initial bacterial concentration of $10^7$ CFU/ml (high load) (Table 1 in S1 File). Finally, when *S. epidermidis* was grown on coupons under limited oxygen availability, biofilm formation was also inhibited, while bacterial growth was not impacted (Table 1 in S1 File).

## Discussion

Biofilm formation varied across the different plastic materials tested, with surface roughness emerging as a key factor. Rough-textured PC bag materials like R-PVC-BTHC and EVA were more conducive to biofilm formation than smoother materials. Lower OD values were obtained for S-PVC-BTHC, likely reflecting its smoother texture, which hampers bio-film formation [19]. Indeed, R-PVC-BTHC has previously been shown to better promote biofilm formation compared to its smooth side [19,20]. EVA also appeared to support biofilm growth, which may be explained by its distinctive porous and irregular texture [22]. PVC-TEHTM bags, with their diamond-shaped textured surface [22], also seemed to support biofilm formation, but to a lesser extent in TSB, potentially influenced by the nature of the plasticizer. However, fewer differences than expected were observed between plastics in PCs, possibly because platelets enhance biofilm adhesion regardless of the plastic type by serving as scaffolds. This interpretation is supported by previous findings showing that platelet components can promote bacterial aggregation and biofilm development in blood products [32,33,35,36]. As previously demonstrated [19,37,38], minimizing the surface roughness of PC bags can be an effective strategy for reducing biofilm formation during PC storage.

Differences observed between the tested growth media may be attributed to their protein content and the presence of platelets. In accordance with the ISO protocol, blank values were subtracted to isolate biofilm-specific signals. In PCs, higher background absorbance, likely due to plasma proteins and platelet aggregates, leads to greater subtraction, which may partly explain the lower biofilm values observed compared to TSB.

While several studies have reported that plasma can promote *S. epidermidis* biofilm formation by providing adhesion-facilitating proteins such as fibrinogen and fibronectin [39,40], our findings show a complete absence of biofilm in plasma under the tested conditions. This discrepancy may be explained by the dual nature of plasma components. In addition to adhesion-promoting proteins, plasma contains antimicrobial peptides and neutralizing antibodies that can inhibit bacterial growth and limit biofilm formation [31]. These molecules interact with bacterial membranes, increasing their permeability and leading to bacterial death [41], while antibodies may block key biofilm-associated proteins such as the accumulation-associated protein [42]. In our study, bacterial viability in plasma was significantly reduced, suggesting that these inhibitory effects predominated. Another important factor to consider is the absence of platelets in plasma, which could limit bacterial adhesion and biofilm development. In contrast, biofilm formation was consistently observed in PCs, despite their suspension in 100% plasma. This suggests that platelets may counteract the antimicrobial effects of plasma by promoting bacterial adhesion and protecting bacteria from host defenses [32,33], underscoring their pivotal role in biofilm formation.

Several factors may account for the limited biofilm formation when culturing *S. epidermidis* directly in the bags. Firstly, continuous agitation at 70 cycles per minute, which simulates the storage conditions of PCs, may hinder the initial bacterial attachment to the plastic surfaces. A similar observation was made with plates in TSB (S2 Fig), indicating that a high agitation speed could help prevent bacterial adhesion.

Secondly, although PC storage conditions are considered aerobic [43], oxygen availability may differ between PC storage and plates due to differences in gas exchange mechanisms. While PC bags are sealed and rely on gas diffusion through the plastic, polystyrene plates are loosely covered, allowing passive gas exchange with ambient air and potentially greater oxygen accessibility. Our supplementary experiments (S1 File) showed that biofilm formation was inhibited under low oxygen conditions, while bacterial growth remained unaffected, indirectly supporting the idea that oxygen availability may influence biofilm development. Although agitation in PC bag storage conditions may enhance mixing, it does not compensate for the limited surface area available for gas diffusion. Future studies should include direct measurements of oxygen concentration in both systems to validate this hypothesis. While *S. epidermidis* can form biofilms under aerobic conditions by creating anaerobic microenvironments within the biofilm itself [44,45], external oxygen availability may still influence the initiation and extent of biofilm development.

Thirdly, differences in surface-to-volume ratios between the coupon and bag experiments may help explain the observed results. In the case of coupons, the surface-to-volume ratio for biofilm formation is 1.25 cm²/ml, and the entire surface of the coupons is analyzed. For bags, the surface-to-volume ratio of the analyzed area ranges from 0.1 to 0.9 cm²/ml, depending on the PC bag type, with only 9% to 28% of the total bag surface area assessed. This significant difference in surface-to-volume ratios could impact biofilm formation by influencing bacterial deposition on the plastic. The smaller area analyzed for bags decreases the likelihood of encountering sections with biofilm. The variability in these ratios highlights the limitation of the ISO 4768:2023(E) standard, which is designed for static analyses on plastic coupons and does not accurately reflect storage conditions for complete PC bags.

A key limitation of the CV assay is its inability to quantify floating aggregates, potentially biofilm-like structures, that were predominantly observed in the inoculated PCs and TSB within PC storage bags. Because these aggregates are not attached to the plastic surfaces, they fall outside the scope of CV-based quantification, consequently resulting in an incomplete assessment of biofilm presence. This limitation is specific to floating aggregates bacteria, such as *S. epidermidis* ATCC 35984. However, this strain is recommended by the ISO 4768:2023(E) standard for testing biofilm formation on plastic materials and is widely used in studies on biofilms in blood products and antibacterial surface properties

[6,19,31–33,38]. Future studies using transfusion-relevant isolates that form surface-attached biofilms could help overcome this limitation and provide a more comprehensive evaluation of biofilm formation in PC storage bags.

Blood banks are continually seeking new methods to prevent the risk of contamination in blood products. Blood devices manufacturers use the EVA plastic in combination with platelet additive solutions (PAS). The transparency of EVA to UV radiation allows its use for PRTs while PAS supplies essential nutrients to platelets. The addition of PAS could have an impact on biofilm contamination of PCs. Studies have shown that biofilm formation by bacteria such as *S. epidermidis* and *S. liquefaciens* is generally reduced in PCs stored in PAS compared to those stored in plasma [46]. This reduction could be attributed to the composition of PAS, which lacks glucose, an essential nutrient for many bacterial species. Although residual plasma in PAS-stored PCs provides sufficient glucose for platelets, the overall concentration remains significantly lower during storage, potentially limiting bacterial proliferation and biofilm development [47]. Further studies are needed to investigate the impact of PAS, EVA-based device materials and PRTs on the biofilm-forming behavior of bacterial strains typically contaminating PCs.

Because the sample size of this study was relatively small (n = 3) and the results were specific to the strain of *S. epidermidis* ATCC 35984, they highlighted the need for future research to explore the implications for other bacterial species. Additionally, while the CV assay is effective, it could benefit from refinement to enhance its sensitivity and detection capabilities for biofilm analysis. Overall, this study represents a significant step toward developing improved methods for characterizing biofilm formation in blood products.

Using the ISO 4768:2023(E) standardized protocol, this study has demonstrated that the texture and composition of plastics used in blood product devices can influence the ability of contaminants to form biofilms. Although additional work is needed to determine whether these differences may have implications for the prevention of biofilm-related TTBIs, the near-absence of biofilm in PC bags is reassuring. This suggests that the PC storage conditions in blood banks are not favorable to *S. epidermidis* ATCC 35984 biofilm formation, thereby reducing the risk of undetected bacterial contamination.

## Supporting information

**S1 Fig. Schematic representation of the platelet concentrate storage bags used in this study.** Two formats are shown: full-size bags, used for experiments involving TSB and plasma, and reduced-volume bags, created by heat-sealing full-size bags, used for experiments with PCs. This figure illustrates the physical dimensions of the PVC-BTHC (A, D), PVC-TEHTM (B, E) and EVA (C, F) bags (cm), the internal surface areas available for gas exchange (cm² for surface areas), and the total volume (ml) after the addition of the tested storage medium. Specific regions where 2.5 × 2.5 cm coupons were cut for CV experiments are shown on the reduced-volume PC bags (D, E, F) and the full-size PC bags (G, H, I).
(TIF)

**S1 File. Additional tests in TSB to assess the impact of agitation and oxygen availability on biofilm formation in platelet concentrate (PC) bags.** Biofilm quantification (OD) and bacterial counts from *S. epidermidis* assays under different incubation conditions: agitation, reduced oxygen, and static incubation at varying concentrations (see Table 1 and S2 Fig).
(DOCX)

**S2 Fig. Biofilm formation and viability of *S. epidermidis* ATCC 35984 according to incubation method.** Mean OD by CV assay (A, C) and mean log bacterial concentration (log CFU/ml) (B, D) are shown for day 2, 5, and 7 after incubation in bags without agitation (A, B) and on coupons with agitation (C, D) in TSB (n = 1). R-PVC-BTHC (·-●-·), S-PVC-BTHC (━■━), PVC-TEHTM (━◆━) and EVA (·-▲-·) are represented.
(TIF)

## Acknowledgments

Medical writing assistance was provided by Samuel Rochette, who is an employee of Héma-Québec. Thanks to the microscopy service at Institut de Biologie Intégrative et des Systèmes for the SEM images of the plastics.

## Author contributions

**Conceptualization:** Sahra Fonseca, Jonathan Robidoux, Marie-Pierre Cayer, Danny Brouard.

**Formal analysis:** Jolianne Matte, Sahra Fonseca.

**Funding acquisition:** Jolianne Matte, Marie-Pierre Cayer, Danny Brouard.

**Investigation:** Jolianne Matte, Sahra Fonseca, Jonathan Robidoux.

**Methodology:** Jolianne Matte, Sahra Fonseca, Marie-Pierre Cayer.

**Supervision:** Marie-Pierre Cayer, Danny Brouard.

**Writing – original draft:** Jolianne Matte.

**Writing – review & editing:** Jolianne Matte, Sahra Fonseca, Jonathan Robidoux, Steve J. Charette, Marie-Pierre Cayer, Danny Brouard.

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
