## [Decision Letter · Decision Letter 0]

19 May 2025

Dear Dr. Cayer,

Thank you for submitting your manuscript to PLOS ONE. After careful consideration, we feel that it has merit but does not fully meet PLOS ONE’s publication criteria as it currently stands. Therefore, we invite you to submit a revised version of the manuscript that addresses the points raised during the review process.

We look forward to receiving your revised manuscript.

Kind regards,

Geelsu Hwang, Ph.D.

Academic Editor

PLOS ONE

Journal Requirements:

3. We notice that your supplementary figures are uploaded with the file type 'Figure'. Please amend the file type to 'Supporting Information'. Please ensure that each Supporting Information file has a legend listed in the manuscript after the references list.

4. Please remove all personal information, ensure that the data shared are in accordance with participant consent, and re-upload a fully anonymized data set.

Reviewers' comments:

Reviewer's Responses to Questions

**Comments to the Author**

1. Is the manuscript technically sound, and do the data support the conclusions?

Reviewer #1: Partly

Reviewer #2: Yes

2. Has the statistical analysis been performed appropriately and rigorously?

Reviewer #1: N/A

Reviewer #2: Yes

3. Have the authors made all data underlying the findings in their manuscript fully available?

Reviewer #1: No

Reviewer #2: Yes

4. Is the manuscript presented in an intelligible fashion and written in standard English?

Reviewer #1: No

Reviewer #2: Yes

Reviewer #1: The manuscript attempts to analyze the biofilm-forming capacity of Staphylococcus epidermidis ATCC 35984 on four different platelet concentrate (PC) storage bag materials. While the topic is relevant to the role of biofilms in transfusion safety, the study lacks the data quality, depth, and clarity necessary for publication in PLOS ONE.

1. Insufficient Data Quality and Depth:

The study does not present growth kinetics of S. epidermidis under the tested conditions and only reports biofilm formation. Without corresponding planktonic growth data, it is difficult to interpret the biofilm results.

2. Clarity of Writing:

The manuscript's writing is often unclear, making it challenging to follow the methodology and results. Major revisions are required for clarity and flow.

3. Experimental Design Limitations:

Biofilm formation is expected to differ between TSB, plasma, and PC.

It's unclear how long S. epidermidis was cultured in TSB before inoculation into plasma or PCs, or whether inoculation was performed from TSB culture or glycerol stock. This information is critical.

Ethical Committee Clarification (Line 101):

Please specify the number. “Héma-Québec's ethics committee”

4. Misplaced or Ambiguous Text:

Line 122: Reference 25 is included as a full title in the main text. This should be rephrased or removed.

Abstract: It states that four materials were tested, but only two (PVC and EVA) are listed. Please clarify.

Line 139: "At a maximum" it should specify "OD600" for bacterial growth.

Line 152: Clarify whether bacteria were inoculated into plasma or platelets, and from what source (TSB, glycerol stock).

Line 159: Remove double brackets: “([20–24 °C])”.

5. Figures and Data Presentation:

Figure 1B: The image is not clear and needs improvement.

Figure 2E: The X-axis labeling is inadequate; values are not fully visible.

Initial Log(CFU/mL) is different in TSB and PCs.

Figure 4: It is unclear how presenting biofilm formation as area under the curve (AUC) adds new insight. This seems repetitive of prior data.

Authors should provide images of the PC coupons and storage bags used.

Additional Clarifications Required:

• Statistical analysis: Ensure p-values or effect sizes are reported for key comparisons.

• Line 132: Authors mention 2.5 × 2.5 cm coupons. What was the size of the full PC storage bags used?

• Line 140: Confirm whether the culture was adjusted to 10⁴ CFU/ml in all conditions (TSB, PC, plasma).

Language Corrections:

For example, "Significative" → "significant"

Reviewer #2: Reviewer Comments

• The work is interesting however there are few suggestions for the improvement of the manuscript.

• The abstract says that - The four materials tested included plasticized polyvinyl chloride (PVC) and ethylene vinyl acetate (EVA) bags. To the reader it would appear that only two materials are mentioned here. Though the materials are only two ie PVC and EVA, however you have tested with PVC- BTHC-Smooth, PVC- BTHC-Rough, PVC-TEHTM and EVA. So, it would be better to clearly mention this in the abstract.

• It is important to mention the following in the abstract otherwise it would be confusing for the reader:

o Coupons and bags of each of the materials have been used.

o The PVC-BTHC coupons have both a smooth and a rough side.

• Abstract: … rough side of PVC-BTHC showed significative biofilm formation … / Please correct as … rough side of PVC-BTHC showed significant biofilm formation.

• Methods: Preparation of the plastic coupons could be given as a separate section or included in the PC bags section as PC bags and coupons.

• Results and Discussion: Though the authors have explained in the manuscript why there is no biofilm formation when Staphylococcus epidermidis is grown in plasma with coupons literature does reveal that plasma is a key regulator of S. epidermidis biofilm formation. Therefore, this aspect should be justified clearly.

**Do you want your identity to be public for this peer review?** For information about this choice, including consent withdrawal, please see our Privacy Policy

Reviewer #1: **Yes: ** Basit Yousuf

Reviewer #2: No

---

## [Author Response · Author response to Decision Letter 1]

25 Jun 2025

Please refer to the attached response document.

---

## [Decision Letter · Decision Letter 1]

5 Aug 2025

Dear Dr. Cayer,

Thank you for submitting your manuscript to PLOS ONE. After careful consideration, we feel that it has merit but does not fully meet PLOS ONE’s publication criteria as it currently stands. Therefore, we invite you to submit a revised version of the manuscript that addresses the points raised during the review process.

We look forward to receiving your revised manuscript.

Kind regards,

Geelsu Hwang, Ph.D.

Academic Editor

PLOS ONE

Journal Requirements:

Reviewers' comments:

Reviewer's Responses to Questions

**Comments to the Author**

Reviewer #2: All comments have been addressed

Reviewer #3: (No Response)

Reviewer #4: (No Response)

2. Is the manuscript technically sound, and do the data support the conclusions?

Reviewer #2: (No Response)

Reviewer #3: Partly

Reviewer #4: Yes

3. Has the statistical analysis been performed appropriately and rigorously?

Reviewer #2: (No Response)

Reviewer #3: I Don't Know

Reviewer #4: Yes

4. Have the authors made all data underlying the findings in their manuscript fully available?

Reviewer #2: (No Response)

Reviewer #3: Yes

Reviewer #4: Yes

5. Is the manuscript presented in an intelligible fashion and written in standard English?

Reviewer #2: (No Response)

Reviewer #3: Yes

Reviewer #4: Yes

Reviewer #2: (No Response)

Reviewer #3: Matte and colleagues designed a protocol to compare S. epidermidis biofilm formation on platelet bags made of different plastic materials. The experimental design did not include transfusion relevant bacterial strains, which needs to be justified. Strain ATCC 35984 is a strong biofilm former, difficult to manipulate, which makes results interpretation challenging. Another concern is the use of the ATCC 12228 strain as a negative control in experiments involving PCs when it is known that this strain converts to a biofilm positive phenotype when grown in PCs. The authors should present results of biofilm formation on coupons with ATCC 12228 to justify using it as a threshold. The discussion needs improvement to demonstrate the novelty and how the results presented herein advance knowledge. There are other studies which have shown that rough surfaces of PC bags promote biofilm formation by bacteria. The use of the ISO 4768:2023(E) standard is commended as it allows for assay standardization; however, the authors could comment on how this method differs from those previously published.

SPECIFIC COMMENTS

ABSTRACT

1. Line 28. PCs are stored at "20-24 C" is preferably than "room temperature" as ambient temperature can change in different locations.

2. Line 29. The statement “making those products susceptible to contamination” is inaccurate as PC storage at room temperature does not make the product more susceptible to contamination, it potentially favors growth of bacteria already introduced during venipuncture.

3. Line 48. From the results presented (Line 43: “similar biofilm amount across PC bag materials in the presence of platelets”), it is not clear how the following conclusion can be made “However, platelet bag materials limit biofilm formation under standard blood bank storage conditions, reducing the risk of undetected bacterial contamination”. Please revise.

INTRODUCTION

1. Line 52. Please see comment above about “room temperature”

2. Line 53. “Consequently, PCs are more at risk of contamination” is incorrect. Please clarify to state that PCs are more CONDUCIVE to proliferation of bacteria introduced during blood donation due to their storage conditions.

3. Line 62. References 11-14 support the incidence of TTBIs post-implementation of culture systems not pathogen reduction. The safety risk of PCs screened with culture methods or treated with pathogen reduction technologies is not the same and should not be combined as one category.

MATERIALS AND METHODS

1. How many biological repetitions of the experiments were done?

2. Line 126. Why was TSB not supplemented with 0.5-1.0% of glucose? It is well known that glucose supplementation of TSB increases biofilm formation by staphylococci.

3. Line 130. Which type of plates were used to test PC bags’ coupons? How was bacterial attachment directly to the plates avoided? Was interference of attachment to the plate with attachment to the coupons considered?

4. Line 131. Were PCs stored in a platelet incubator? If not, please provide storage conditions (temperature and agitation).

5. Line 145. How was estimation of bacterial load adjusted without colony counts?

6. Line 149. Please elaborate in your statement “Variations in the initial bacterial load may have occurred due to the inherent variability of spectrophotometric estimation” Was the variability estimated? It is known that working with a robust biofilm former strain such as ATCC 35984 is challenging as bacterial aggregates are difficult to resuspend resulting in great variability of bacterial counts. Why were not biofilm positive and negative PC isolates of S. epidermidis used instead of commercial strains?

7. Line 157. Why were the coupons incubated at 22+/- 1C? Why were they incubated with no agitation?

8. Line 171. How many times the coupons were washed?

9. Line 189. Biofilm-negative staphylococcal strains convert to a biofilm positive phenotype when grown in PCs, why was the ATCC 12228 strain then used as a negative control for biofilm formation assays in platelet products?

RESULTS

1. Lines 241 and 282. Please replace “concentration” with “load”

2. Line 293. Regarding the statement “Biofilm formation on coupons between TSB and PCs was compared for each type of plastic and each day of analysis; however, no statistical differences were observed”, please provide the p values for each day between TSB and PCs.

3. Figure 2 shows differences between media and PCs. Even if there are no statistical differences, biologically, what is the meaning of having values within the threshold zone for some samples and not others? It is concerning that biofilm formation results with the ATCC 12228 strain were used to establish a threshold when this strain is known to form biofilms in PCs and therefore can tamper the interpretation of the results. Non-spiked PCs can be used as baseline for biofilm formation assays in this milieu.

4. Please define asterisks in figure legends.

Discussion

1. Line 324. Please explain your statement “No biofilm was observed in plasma, which may be partly due to plasma's antimicrobial peptides and neutralizing antibodies that limit bacterial growth and, consequently, S. epidermidis biofilm formation”. PCs used in the study were suspended in 100% plasma and therefore they also had antimicrobial peptides and antibodies. Please consider discussing the absence of platelets in plasma, which could a the limiting factor for biofilm formation.

2. Line 326. Please explain what the difference in gas accessibility between plates and PC bags is. PC containers are made of gas permeable material and are stored under agitation. One would expect that there is more oxygen accessibility in the bags compared to the coupons stored in plates made of hard plastic.

3. Line 341. For the statement “While several studies have reported that plasma can promote S. epidermidis biofilm formation by providing adhesion facilitating proteins such as fibrinogen and fibronectin, our findings show a complete absence of biofilm in plasma under the tested conditions.” Please provide supporting references as the interpretation of published literature may be inaccurate. While plasma proteins promote bacterial adhesion, mature biofilms are formed in the presence of platelets.

4. Lines 345-347. Please see comment above on “antimicrobial peptides and neutralizing antibodies” which are also contained in PCs.

5. Line 362. Reference 36 does not support the statement “oxygen availability may be lower in PC storage bags than in plates, leading to reduced biofilm development”. Furthermore, the statement is speculative, the authors would need to measure oxygen content in a static plastic culture plate and demonstrate that is higher than in a gas permeable PC bag under agitation.

6. Line 375. This limitation could be omitted if the experiments of this study are repeated/validated with transfusion relevant isolates that form biofilms but do not form floating aggregates such as the ones produced by strain ATCC35984.

7. Line 387. PCs produced in PAS may have reduced glucose levels, which the authors could calculate, but glucose is still present.

8. There is no discussion about PC treatment with pathogen reduction technologies, which needs to be taken into consideration in the context of biofilm formation.

9. How different is the protocol of ISO 4768:2023(E) standard compared to published literature?

10. There is a lack of discussion of the results obtained in the study compared to published literature (e.g., https://pubmed.ncbi.nlm.nih.gov/28961393/;
https://pubmed.ncbi.nlm.nih.gov/30354054/;
https://pubmed.ncbi.nlm.nih.gov/27554133/) What is the novelty of this study?

11. There is a lack of discussion of the results of this study for biofilm formation in TSB vs PCs in relation to previous studies which have shown that biofilm formation by S. epidermidis is enhanced in PCs compared to TSB supplemented with glucose. The results presented herein showed the opposite and a potential explanation is needed.

12. Please present results of the ATCC12228 strain in all conditions. Previous studies have shown positive biofilm formation by this strain in PCs. If biofilm formation results with this strain were used as threshold (line 196), this may explain the reduction in biofilm formation on coupons in PCs compared to TSB and this needs to be discussed.

Reviewer #4: The authors have compared the formation of biofilms by Staphylococcus epidermidis on standard-sized coupons from four commonly used platelet concentrate (PC) storage packs, as well as whole packs stored for up to seven days under standard conditions of ambient temperature with constant horizontal agitation. The potential for bacterial contamination of PC under these storage conditions is well known. Although the sample size is acknowledged to be small, the results are suitably controlled, and the paper adds to the body of knowledge aiming to reduce this transfusion hazard.

The paper is much improved following the detailed responses from the authors to the comprehensive comments from the previous reviewers. As such, this reviewer has only some minor comments, detailed below.

Materials and methods

Supplementary Figure S1: The volume and surface area displayed for each pack pair (for example, A and D) are the same. Should this be the case, considering that the pictures D to F denote a reduced volume pack?

Line 204: "The resulted suspensions…" Suggest this is changed to "resulting" or "resultant" suspensions…

Results

Captions for figures 2, 3 and 4: Please make clear which measure of variance is being used for the error bars.

**Do you want your identity to be public for this peer review?** For information about this choice, including consent withdrawal, please see our Privacy Policy

Reviewer #2: No

Reviewer #3: No

Reviewer #4: No

---

## [Author Response · Author response to Decision Letter 2]

2 Sep 2025

We have addressed all reviewer and editor comments in detail in a separate response document, which is included with our submission.

---

## [Decision Letter · Decision Letter 2]

15 Sep 2025

Monitoring of *Staphylococcus epidermidis*  biofilm formation on platelet storage bag surfaces

PONE-D-25-18826R2

Dear Dr. Cayer,

We’re pleased to inform you that your manuscript has been judged scientifically suitable for publication and will be formally accepted for publication once it meets all outstanding technical requirements.

Kind regards,

Geelsu Hwang, Ph.D.

Academic Editor

PLOS ONE

Reviewers' comments:

Reviewer's Responses to Questions

**Comments to the Author**

Reviewer #3: All comments have been addressed

Reviewer #4: All comments have been addressed

2. Is the manuscript technically sound, and do the data support the conclusions?

Reviewer #3: Yes

Reviewer #4: (No Response)

3. Has the statistical analysis been performed appropriately and rigorously?

Reviewer #3: Yes

Reviewer #4: (No Response)

4. Have the authors made all data underlying the findings in their manuscript fully available?

Reviewer #3: Yes

Reviewer #4: (No Response)

5. Is the manuscript presented in an intelligible fashion and written in standard English?

Reviewer #3: Yes

Reviewer #4: (No Response)

Reviewer #3: The authors have adequately addressed the Reviewers' comments and have provided rationale for chosen approaches in the experimental design.

Reviewer #4: (No Response)

**Do you want your identity to be public for this peer review?** For information about this choice, including consent withdrawal, please see our Privacy Policy

Reviewer #3: No

Reviewer #4: No

---

## [Editor Report · Acceptance letter]

PONE-D-25-18826R2

PLOS ONE

Dear Dr. Cayer,

I'm pleased to inform you that your manuscript has been deemed suitable for publication in PLOS ONE. Congratulations! Your manuscript is now being handed over to our production team.

Kind regards,

on behalf of

Dr. Geelsu Hwang

Academic Editor

PLOS ONE